
# Towards understanding the mean annual water-energy balance equation based on an Ohms-type approach

Xu Shan[1], Xingdong Li[1], Hanbo Yang[1]

[1]State Key Laboratory of Hydro-Science and Engineering, Department of Hydraulic Engineering, Tsinghua University,
Beijing 100084, China

*Correspondence to*: Hanbo Yang (yanghanbo@tsinghua.edu.cn)

**Abstract.** The Budyko hypothesis has been widely used to describe precipitation partitioning at the catchment scale. Many empirical and analytical formulas have been proposed to describe the Budyko hypothesis. Based on dimensional analysis and mathematic reasoning, previous studies gave an analytical derivation, i.e., the Mezentsev-Choudhury-Yang (MCY) equation. However, few hydrological processes were involved in the derivation. Therefore, this study firstly defines a catchment network to describe water vapor transformation and transportation using the Lagrangian particle tracking method; and then proposes the generalized flux of water vapor, which can be expressed as the ratio of potential difference with resistance. Furthermore, this study obtains a new constraint for the mean annual water-energy balance, $\frac{1}{f(E)} = \frac{1}{f(E_0)} + \frac{1}{f(P)}$ with $E$, $E_0$ and $P$ being evaporation, potential evaporation and precipitation, respectively, and $f(\ )$ being a function of generalized flux, based on an analogy of the Ohms-type approach and the homogeneity assumption, i.e., the generalized flux has the same form for both water vapor transportation and chase transformation, and in other words, precipitation and potential evaporation have an equalized effect on evaporation. According to this constraint, the MCY equation can be obtained when the generalized flux $f(\ )$ is a power function. In addition, this study suggests a more general expression $E = \frac{P(b+kE_0)}{[P^n+(b+kE_0)^n]^{1/n}}$ under conditions without the homogeneity constraint, where $E$, $E_0$ and $P$ are evaporation, potential evaporation and precipitation, respectively, and $n$, $k$ and $b$ are constants (MCY equation when $b = 0$ and $k = 1$).

## 1 Introduction

The mean annual water-energy balance equation describes the long-term relationship of actual evaporation ($E$) with precipitation ($P$) and potential evaporation ($E_0$) at the catchment scale. This equation is widely used in ecological, climatological, and socioeconomic applications (Greve et al., 2015). Additionally, this equation has been proved to be a powerful tool to assess changes in catchment water balance as a function of climate change (Roderick and Farquhar, 2011; Yang and Yang, 2011; Renner et al., 2012; van der Velde et al.,2013; Greve et al., 2015).

Many attempts were made to formulate the mean annual water-energy balance according to observations from different catchments (Schreiber, 1904; Ol'dekop, 1911; Budyko, 1958; Pike, 1964). Based on previous studies, Budyko (1974) proposed a hypothesis on the mean annual water-energy balance, i.e., the Budyko hypothesis, which was expressed mathematically as follows:





$$E = E(E_0, P),\tag{1}$$

with the boundary conditions:

$$E \to P \text{ as } E_0 \to \infty$$

$$E \to E_0 \text{ as } P \to \infty,\tag{2}$$

which are commonly referred to as "dry condition" and "wet condition". Initially, the function was suggested without any parameters, indicating no capacity to control the impact of different catchment characteristics on the water-energy balance. Later, considering the effects of landscape characteristics, an adjustable parameter was introduced to describe the impacts of catchment characteristics on the water-energy balance (Choudhury, 1999).

In addition, many studies have attempted to achieve an analytical equation based on mathematical reasoning. First, Bagrov

(1953) introduced a derivative of the mean annual water-energy balance, $dE/dP=1-(E/E_0)^n$, and Mezentsev (1955) assumed $m=(n+1)/n$, giving a modification of $dE/dP=[1-(E/E_0)^n]^m$ and obtaining an integration of

$$E = PE_0/(P^n + E_0^n)^{1/n} .\tag{3}$$

However, the meaning of $m=(n+1)/n$ was not given by Mezentsev (1955). Then, Fu (1981) assumed that the derivative of $E$ with respect to $P$ (or $E_0$) could be expressed as a function of the variables $E_0 - E$ and $P$ (or $P - E$ and $E_0$), i.e.,

$$\frac{\partial E}{\partial P} = f(E_0 - E, P),\tag{4}$$

$$\frac{\partial E}{\partial E_0} = f(P - E, E_0).\tag{5}$$

Furthermore, he derived one analytical solution by dimensional analysis and mathematical reasoning (Fu, 1981; Zhang et al., 2004) as follows:

$$\frac{E}{P} = 1 + \frac{E_0}{P} - [\left(1 + (\frac{E_0}{P})^w\right)]^{1/w},\tag{6}$$

Yang et al. (2008) suggested a more general assumption that $E$ can be described as an implicit function of $P$, $E_0$ and $E$, i.e., $E=E(P, E_0, E)$ (equation (5) in Yang et al., 2008), together with the boundary conditions, namely, a 0-order boundary condition similar to equation (2) and a 1-order boundary condition as follows:

$$\begin{cases} \dfrac{\partial E}{\partial P} = 0, & \text{at } P/E_0 \to \infty, \text{ or } E = E_0 \\[2mm] \dfrac{\partial E}{\partial E_0} = 0, & \text{at } E_0/P \to \infty, \text{ or } E = P \\[2mm] \dfrac{\partial E}{\partial P} = 1, & \text{at } P \to 0, E_0 \neq 0 \\[2mm] \dfrac{\partial E}{\partial E_0} = 1, & \text{at } E_0 \to 0, P \neq 0 \end{cases},\tag{7}$$





Furthermore, Yang et al. (2008) analytically derived a solution for the Budyko hypothesis that was similar to the formula derived by Mezentsev (1955) and suggested by Choudhury (1999) (equation (3)).Thus equation (3) was therefore called the Mezentsev-Choudhury-Yang (MCY) equation. Recently, Zhou et al. (2015) gave a general derivation of all kinds of Budyko functions by introducing a generator function:

$$g(\varphi) = \frac{\frac{\partial E}{\partial P}}{\frac{\partial E}{\partial E_0}} \frac{P}{E_0} = \frac{F(\varphi) - \varphi F'(\varphi)}{\varphi F'(\varphi)}, \tag{8}$$

where $\Phi = E_0/P$ and $F(\Phi) = E/P$. Then, they obtained the MCY equation by choosing $g(\Phi) = \Phi^n$ and solving equation (8). However, no more hydrological explanation of $g(\varphi)$ or MCY equation were given. Notably, the differential equations proposed in those studies are not very rigorous and do not reflect sufficient hydrological understanding.

Regarding to physics meaning, the hydrological cycle shapes energy balances and interacts strongly with atmospheric
motion and transport (Kleidon et al, 2013). Fluxes displayed in the hydrological cycle, such as evaporation and precipitation, could be described by thermodynamics. Accordingly, thermodynamic principles, such as the principle of maximum entropy production (MEP) (McDonnell et al., 2007; Kleidon and Schymanski, 2008; Kleidon, 2009, 2010 a,b; Zehe and Sivapalan, 2009; Schaefli et al., 2011) and Carrot Limit (Kleidon et al, 2013), are widely used to understand the hydrological cycle. Kleidon and Schymanski (2008) reviewed the hydrological applications of MEP and proposed the expressions for entropy
production. Wang et al. (2015) introduced their expressions to study catchment water balance and developed a two-parameter equation approaching the Budyko hypothesis, as follows:

$$\frac{E}{P} = \frac{1 + \varphi\varepsilon - \varepsilon + \varphi\frac{E_0}{P}\sqrt{\left(1 + \varphi\varepsilon - \varepsilon + \varphi\frac{E_0}{P}\right)^2 - 4\varphi\varepsilon(1 + \varphi - \varepsilon)\frac{E_0}{P}}}{2\varepsilon(1 + \varphi - \varepsilon)}, \tag{9}$$

where $\varepsilon$ represents the initial evaporation ratio and $\varphi$ represents the ratio of the continuing evaporation conductance to the runoff conductance. Zhao et al. (2016) further derived a general catchment water balance expression unifying catchment
water balance equations at different time scales. However, Westhoff et al. (2016) pointed out that the results of Wang et al. (2015) had some contradictions with Westhoff and Zehe (2013).

Accordingly, in this paper, focusing on the subsequent transportation processes of the precipitated water over a certain catchment, we define a catchment network and assume that fluxes (including vapor transportation and phase transition) can be estimated according to an Ohms-type approach. Furthermore, we obtain a physical constraint for the mean annual water-
energy balance. Section 2 gives the basic assumptions and a conceptual framework, which can lead to MCY equation, Section 3 gives the main reasoning, and the discussion and conclusions are given in Section 4 and Section 5, respectively.





## 2 Ohms-type approach

In a catchment, there are two kinds of water phase transition at the mean annual scale, namely evaporation, condensation of the water vapor to precipitation. Water vapor enters a certain catchment through atmospheric motion, and then condenses as precipitation. Part of the liquid water would evaporate as evaporation, the other part of the liquid water will converge as runoff. Subsequently, water vapor from evaporation can be precipitated in the same catchment or transported to other catchments due to atmospheric motion. We assume that the water phase transitions and transportations can be approached using an Ohms-type law at the mean annual scale, which is detailed by definitions and assumptions regarding to flux.

### 2.1 Definitions and assumptions

In a catchment, water vapor condenses to precipitation ($P$), and then, part of the precipitation evaporates ($E$), while runoff ($R$) is formed from the other part of the precipitation. Over a long duration and by ignoring the water storage change, the catchment water balance can be expressed as

$$E = P - R. \tag{10}$$

First, we focus on the phase transition and transportation of water and propose a catchment network at the mean annual scale (Figure 1). As shown in Figure 1, Catchment $A_1$ is a chosen catchment for water balance analysis. $P_1$ is the precipitation falling on Catchment $A_1$. Here, we track the transformation and transportation of $P_1$ by using the Lagrangian particle tracking method. Firstly, $P_1$ partitions into two parts, evaporation $E_1$ and runoff $R_1$. Then, the water vapor corresponding to $E_1$ precipitates on Catchment $A_1$ and other catchments, which are denoted by Catchment $A_{2,j}$ ($j$=1, 2, 3,…). The precipitation originating from $E_1$ and falling on Catchment $A_{2,j}$ is denoted by $P_{2,j}$. The sum of $P_{2,j}$ ($j$=1, 2, 3,…) is denoted by $P_2$. Notably, $P_{2,j}$ is just part of the precipitation falling on Catchment $A_{2,j}$, i.e., not the total precipitation on the catchment. Next, $P_2$ partitions into two parts, evaporation $E_2$ and runoff $R_2$. Similarly, $E_2$ is all the evaporation originating from $P_{2,j}$ ($j$=1, 2, 3,…), and it precipitates on catchments, which are denoted by Catchment $A_{3,j}$ ($j$=1, 2, 3,…). Figure 1 shows that $P_1$ is divided into runoff $R_i$ ($i$=1, 2, ..., $n$) and evaporation $E_n$ in the catchment network. $E_1$ is part of $P_1$, i.e. $E_1 = k_1 P_1$, with $0 < k_1 < 1$. Similarly, $E_2$ is part of $P_2$ ($E_1$), so $E_2 = k_2 P_2 = k_2 E_1 = k_1 k_2 P_1$, with $0 < k_2 < 1$. Finally, $E_n = \prod_{i=1}^{n} k_i P_1$, with $0 < k_i < 1$. Therefore, when $n \to \infty$, there is $E_n \to 0$ and $P_1 = \sum_{i=1}^{n} R_i$. In other words, the initial precipitation $P_1$ (falling on Catchment $A_1$) completely transforms into runoff after numerous evaporation-precipitation transformations.

In this study, the generalized flux is defined as the potential difference divided by the resistance and is a function of flux. That is, all the generalized fluxes here are driven by some kind of potential difference or potential gradient.

In addition, some essential assumptions are given as:

**Assumption 1**: The mathematical form of the generalized flux is a positive single-value increasing function with respect to the absolute amount of water flux within the water movement process during a certain period.

**Assumption 2**: The mathematical form of the generalized flux does not vary with different water movement processes within a catchment and between catchments.





**Assumption 3**: The potential of liquid water is assumed to be zero.

The generalized flux can be defined according to the resistance of the water vapor movement or transportation process $\eta$, i.e.

$$f(x) = \frac{\Delta U}{\eta},\tag{11}$$

where $\Delta U$ represents the potential difference and the generalized flux $f(x)$ represents is a function of flux (such as precipitation, evaporation and runoff, denoted by $x$) in the transportation and phase transition processes.

## 2.2 Physical reasoning

We focus on the precipitation partition over Catchment $A_1$. In Figure 2, Node B represents Catchment $A_1$, and Node A represents the atmosphere over Catchment $A_1$. Catchment $A_2$ represents a group of catchments where the water vapor from $E_1$ can precipitate. Similarly, Node D represents Catchment $A_2$, and Node C represents the atmosphere over Catchment $A_2$. $V$ is the water vapor that precipitates on Catchment $A_1$ (precipitation $P_1$). Over a long duration, the net water vapor flux transported from Node A to Node B equals $P_1-E_1$ ($R_1$), flux from Node A to Node C equals $E_1$, and flux from Node C to Node D equals $P_2-E_2$ ($R_2$) (liquid state). According to the definition, the generalized flux between Nodes A and B is $f(R_1)$, that between Nodes A and C is $f(E_1)$, and that between Nodes C and D is $f(R_2)$.

1) Net water vapor flux is transported into Node A via Path $P_1$ in the form of total precipitation.

2) Water exists in a gas state in Nodes A and C and a liquid state in Nodes B and D. Thus, Path A→B represents the phase transition of vapor in the process of condensation. Path A→C represents the vapor transportation driven by the potential difference $U_2 - U_1$.

3) The potential difference between B and D is zero since the potential of liquid water is zero. The potential difference driving the phase transition of condensation is equal to the potential difference between the vapor and liquid water. The potential difference between A and D ($\Delta U_{AD}$) equals the one between A and B ($\Delta U_{AB}$), since the potentials of B and D are zero.

Two additional corollaries are as follows:

(a) **Corollary 1**: There are similar resistances during Path A→B and Path C→D since they are the chase transition from vapor to liquid. Therefore, $\eta_1$ and $\eta_3$ have similar values when assuming the same temperature between Path A→B and Path C→D, which means:

$$\eta_1 = \eta_3,\tag{12}$$

(b) **Corollary 2**: There are sufficient occurrences of water transportation as $n \to \infty$, which lead to $\eta_{AB} = \eta_{CD}$. Note that $\eta_{AB} \neq \eta_1$. Here, $\eta_{AB}$ is the resistance of all the possible roads between Node A and Node B, including Path A→B and Path A→C→D→B. Similarly, $\eta_{CD}$ is the resistance of all possible roads between Nodes C and D.

Thus, we have a general equation:





$$\eta_{AD} = \eta_{CD} + \eta_2 ,$$ (13)

According to equation (11), the resistances can be estimated as $\eta_{AB} = \frac{\Delta U_{AB}}{f(P_1)}$ and $\eta_{AD} = \frac{\Delta U_{AD}}{f(E_1)}$. Consequently, the equation $\eta_{CD} = \eta_{AB}$ leads to

$$\eta_{CD} = \eta_{AB} = \frac{\Delta U_{AB}}{f(P_1)} ,$$ (14)

Because $\Delta U_{AD} = \Delta U_{AB}$, we can obtain

$$\eta_{AD} = \frac{\Delta U_{AD}}{f(E_1)} = \frac{\Delta U_{AB}}{f(E_1)} ,$$ (15)

According to the boundary condition, $E_1 \to E_0$ and $R_1 \to \infty$ when $P_1 \to \infty$. This indicates that much more water is draining via Path $R_1$ than evaporating via Path $E_1$, which means $\eta_1 \ll \eta_2$ and $\eta_3 \ll \eta_2$. In addition, the resistance of $\eta_{CD} < \eta_3$ since $\eta_{CD}$ is a result of the parallel of $\eta_3$ and the resistance of the remaining part. Thus, $\eta_2 + \eta_{CD} < \eta_2 + \eta_3$. The boundary

condition $P_1 \to \infty$ yields that $\eta_{AD} = \eta_2 + \eta_{CD} \to \eta_2$, i.e.,

$$\eta_{AD} = \eta_2 ,$$ (16)

Substitution of equation (15) into equation (16) leads to

$$\eta_2 = \frac{\Delta U_{AB}}{f(E_1)} = \frac{\Delta U_{AB}}{f(E_0)} ,$$ (17)

Substitution of equations (14), (15) and (17) into equation (13) leads to

$$\frac{1}{f(E)} = \frac{1}{f(E_0)} + \frac{1}{f(P)} ,$$ (18)

There are the following boundary and limiting conditions for $f(x)$:

$$\begin{cases} f(x) \to 0^+, as\ x \to 0^+ \\ f(x) \to +\infty, as\ x \to +\infty \\ \quad 0 < f(x) < +\infty \\ \quad 0 < f'(x) < +\infty \end{cases}$$ (19)

Thus, a Budyko function can be obtained by using the equation (18) with a specific form of $f(x)$ above and the boundary and limiting conditions in equation (19). Since the only requirement for $f(x)$ is a monotonically increasing function from 0 to ∞,

we can use any appropriate form of $f(x)$ to construct a solution for Budyko Hypothesis. A simple function for $f(x)$ in equation (18), i.e. the generalized function defined in Section 2, is a power function,

$$f(x) = ax^n ,$$ (20)

with $a$ and $n$ being parameters. Then, we can substitute equation (20) into equation (18) and obtain

$$\frac{1}{E^n} = \frac{1}{P^n} + \frac{1}{E_0^n} ,$$ (21)



which can be transformed into the MCY equation, $E = \frac{PE_0}{(P^n + E_0^n)^{1/n}}$.

## 4 Discussions

### 4.1 The generalized flux

Flux is generally defined as the quantity that passes through the surface (Maxwell, 1873). There are several forms of flux,
such as momentum flux ($N \cdot s \cdot m^{-2} \cdot s^{-1}$), heat flux ($J \cdot m^{-2} \cdot s^{-1}$), mass flux ($kg \cdot m^{-2} \cdot s^{-1}$), and electric flux ($N \cdot C^{-1}$). Flux can be
estimated as the potential difference divided by resistance. For example in Darcy's law, the water flux ($Q$) can be estimated
as $Q = J/r$, where $J$ is the hydraulic slope and $r$ is the resistance. In Ohm's law, the electric current ($I$) can be calculated as
$I = U/R$, where $U$ is the electric potential difference, and $R$ is the electric resistance. Notice the similarity in Darcy's law
and Ohm's law, we propose the generalized flux to study the atmospheric motions. Generally the flux is not limited to
linearity. For example, an alternate form of Darcy's law is $v = J/r'$, where $v$ is the velocity (or the flux density) and $r' = r/A$ (where $A$ represents sectional area). In this study, we defined the generalized flux as a function of the flux, i.e., $f(x)$,
where $x$ represents some form of flux. The generalized flux can be used to describe a more general relationship between
fluxes and potential differences. For example, under turbulent conditions, $v^2 + bv = J/K_1$ (Forchheimer, 1901), i.e., the
generalized flux $f(x)=x^2 + bx$. In other words, flux in Ohm's law has a linear relationship with potential difference, while
generalized flux can describe a nonlinear relationship between a given flux and potential difference. Equation (20) defines
the generalized flux of water flux at the catchment scale, and parameter $n$ was reported from 0.4 to 3.8 (with a mean of 1.3)
for 210 catchments across China (Yang et al., 2014). It indicates a nonlinear relationship, except for $n = 1$. In addition, the
mean value of 1.3 is larger than 1, and the catchment water balance is speculated to have some similarity with the behavior
of groundwater flow. Remarkably, some catchments have an $n$ value of less than 1. Therefore, the mechanism behind the
nonlinear relationship needs further study.

The MCY equation can be obtained when the generalized flux is a power function. Besides, another form $f(x) = x^2 + bx$ can
be taken as the generalized flux, similar to (Forchheimer, 1901). It should be noted that any quadratic polynomial can be
expressed as $a(x^2 + bx + c)$, and the coefficient $a$ can be removed when it is substituted into equation (18). In addition, $c = 0$ since $f(x) \to 0^+, as\ x \to 0^+$. The numerical analysis was given in Figure 3, and the results show that $f(x) = x^2 + bx$ (with
$b = 10, 50, 100, 200$) can be approximated as power functions with the determinate coefficient larger than 0.98. Furthermore,
we also approach the cubic polynomial $f(x) = x^3 + bx^2$ ($x/x^3$ can be neglected when $x$ represents $E$ and $P$ with a range of 10-
$10^3$) using power functions, and the determinate coefficient is larger than 0.99, as shown in Figure 4. It means that a
polynomial can be numerically approximated using a power function. Additionally, a quadratic polynomial takes a
disadvantage that its coefficients aren't dimensionless since $x$ is a dimensional variable.

We also perform a global analysis to compare different forms of flux in equation (18), namely the quadratic function ($f(x) = x^2 + bx$) and power function ($f(x) = x^n$). 663 basins across the entire world are chosen for this comparison. Mean annual





potential evaporation and precipitation are from the global dataset GLDAS version 2.0 (available at https://hydro1.gesdisc.eosd is.nasa.gov/data/GLDAS/), while mean annual runoff is from Global Runoff Data Center (GRDC, http://www.grdc.sr.unh.edu/) (Fekete et al.[2002]). Mean annual evaporation is calculated as $E = P - R$ for each basin. An optimal value of the parameters, namely $b$ in the quadratic function ($f(x) = x^2 + bx$) and $n$ in the power function ($f(x) = x^n$), can be inferred through a fitting procedure that minimizes mean absolute errors between modeled evaporation with the measured evaporation. The objective functions follow that

$$n = argmin\left\{[R]_i - [P]_i - \left(\frac{[P]_i}{\left(\left(\frac{[P]_i}{[E_0]_i}\right)^n + 1\right)^{\frac{1}{n}}}\right)\right\} \tag{22}$$

$$b = argmin\left\{[R]_i - [P]_i - \frac{1}{2}\left(-b + \sqrt{b^2 + 4 \times \frac{([E_0]_i^2 + b[E_0]_i)([P]_i^2 + b[P]_i)}{[E_0]_i^2 + b[E_0]_i + [P]_i^2 + b[P]_i}}\right)\right\} \tag{23}$$

After that, we calculate the evaporation as $E_m$ retrospectively, based on different forms of flux in equation (18). Then $E$ and $E_m$ of global basins are scattered in Figure 5. It is clear that power function is a better proxy of flux $f(x)$ than quadratic function, reducing the RMSE from 122.3 mm/a to 43.9 mm/a while increasing $R^2$ from 0.81 to 0.99. Quadratic function $f(x) = x^2 + bx$ underestimates $E$ when it goes to a large value. Since $\sum_{i=0}^{m}(a_i x^i)$ is close to $x^m$ when $x$ approaches to a large value, power functions have a potential to approach the catchment water-energy balance with various characteristics.

### 4.2 Physical understanding of the Budyko hypothesis

According to the Ohms-type approach, the partition of precipitation into evaporation and runoff is dependent on the two resistances $\eta_1$ and $\eta_2$. Resistance $\eta_1$ is related to the condensation processes of water vapor, and it can be estimated as $\eta_1 = \Delta U_{AB}/f(P - E)$. In this study, we assumed that the potential of liquid water is zero, so the potential of water vapor is $\lambda$ under the simplest condition ($n = 1$), and $\eta_1 = \lambda/a$; whereas when $n$ does not equal 1, $\eta_1$ has a sophisticated form similar to Darcy's law under turbulent conditions. Additionally, resistance $\eta_2$ can be estimated as $\eta_2 = \Delta U_{AB}/f(E_0)$ according to equation (11). Remarkably, there is an implicit assumption that $f(x)$ is homogeneous in the horizontal and vertical directions, i.e., the generalized flux has the same form for both water vapor transportation and chase transformation. If $f(x)$ is not homogeneous, we denote $\varphi(E_0) = \Delta U_{AB}/\eta_2$, and we can speculate $\varphi(x) = b + kf(x)$ (where $b$ and $k$ are constants) since $\varphi(x)$ should have the same dimension as $f(x)$. Thus, $\frac{1}{E^n} = \frac{1}{P^n} + \frac{1}{(b+kE_0)^n}$, i.e.,

$$E = \frac{P(b+kE_0)}{[P^n + (b+kE_0)^n]^{1/n}}, \tag{21}$$

When $b = 0$, equation (21) can be simplified as $E = \frac{kPE_0}{[P^n + (kE_0)^n]^{1/n}}$, which is the same as that proposed by Zhou et al. (2015) (equation (21) in their paper). Furthermore, when $b = 0$, $k = 2$, and $n = 1$, it can be transformed into $E = \frac{2PE_0}{P+2E_0}$, which was firstly proposed by Sharif et al. (2007).





This study proposes a catchment network in which the initial water vapor precipitated over Catchment $A_1$ can be completely transformed into runoff after infinite iterations of the evaporation-precipitation process. In the Ohms-type approach, as shown in Figure 2, we assume that the generalized flux $f(x) = ax^n$ has the same values of $a$ and $n$ for Catchments $A_1$ and $A_2$. As is well known, $n$ represents the catchment characteristics (Yang et al., 2008). Therefore, this assumption indicates
Catchments $A_1$ and $A_2$ have similar characteristics. Under the condition that Catchments $A_1$ and $A_2$ have a relatively large difference in catchment characteristics, a large difference in $n$ occurs, which will leads to a more complicated form for the mean annual water-energy balance equation. Therefore, further study on the Ohms-type approach is still required.

Meanwhile a new constraint, equation (18), is given by this study. Previously, the mean annual water-energy balance was only constrained by the 0 order and 1 order boundary conditions (equations (2) and (7)), based on some preliminary
knowledge of dry and wet conditions. There are no more mathematical constraints for the solution, resulting redundant dimensional reasoning and presumptions in previous derivation (Yang et al., 2008). This study gives a new constraint on the mean annual water-energy balance, different solutions would be reached considering different forms of generalized constraints.

## 5 Conclusions

Previous studies have analytically derived the mean annual water-energy balance equation for the Budyko hypothesis mainly by mathematical reasoning, such as Fu (1981), Yang et al. (2008), and Zhou et al. (2015). Towards further understanding on the physical meaning of this equation, this study focuses on subsequent transportation and transformation of the precipitation fallen down to a certain catchment using the Lagrangian particle tracking method, proposes a catchment network in which water vapor is transformed and transported through evaporation-precipitation processes, defines the generalized flux of water
vapor, and expresses the generalized flux as the ratio of potential difference with resistance by using an Ohms-type approach. Based on these reasoning, the relationship among potential evaporation ($E_0$), precipitation ($P$) and evaporation ($E$), $\frac{1}{f(E)} = \frac{1}{f(E_0)} + \frac{1}{f(P)}$, is achieved as a new constraint of mean annual water-energy balance, in which $f(x)$ represents the generalized flux (i.e. a function of flux). Furthermore, the MCY equation $E = \frac{PE_0}{(P^n + E_0^n)^{1/n}}$ is obtained when $f(x)$ is a power function. Remarkably, an implicit homogeneity assumption for the MCY equation is exposed, i.e., the generalize function has the
same form for both vapor transportation and chase transition, and in other words, precipitation and potential evaporation have an equalized effect on evaporation. In addition, without the homogeneity assumption, this study suggest a general form $E = \frac{P(b + kE_0)}{[P^n + (b + kE_0)^n]^{1/n}}$, where $b$ and $k$ are constants. The equation can be simplified to $E = \frac{kE_0 P}{[P^n + (kE_0)^n]^{1/n}}$ proposed by Zhou et al. (2015) if setting $b = 0$; the MCY equation if setting $b = 0$ and $k = 1$; and $E = \frac{2PE_0}{P + 2E_0}$ proposed by Sharif et al. (2007) if setting $b = 0$, $k = 1$ and $n = 1$.




**Acknowledgments**

This research was partially supported by funding from the National Natural Science Foundation of China (Grant Nos. 51622903 and 41661144031), the National Program for Support of Top-notch Young Professionals, and the Program from the State Key Laboratory of Hydro-Science and Engineering of China (Grant No. 2017-KY-01).

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





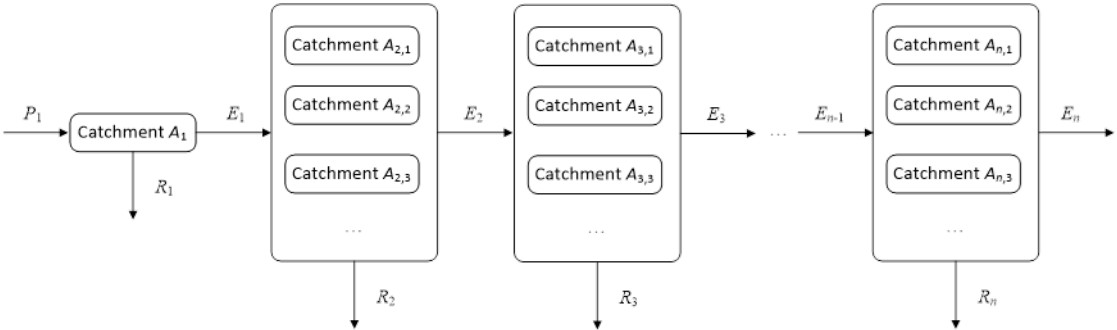

**Figure 1:** A conceptual diagram for the subsequent transportation and transformation of the precipitation fell down to Catchment $A_1$ within the catchment network. Catchment $A_1$ represents the chose catchment for water balance analysis. Catchment $A_{i,j}$ ($i=2, 3, …, n, j=1, 2, 3, …$) represents the catchments that the evaporated water from Catchments $A_{i-1,j}$ ($i=2, 3, …, n, j=1, 2, 3, …$) can reach. $P_1$ denotes the precipitation fell down to Catchment $A_1$. $E_i$ and $R_i$ ($i=1, 3, …, n$) denote the evaporation and runoff only from $P_1$, respectively.





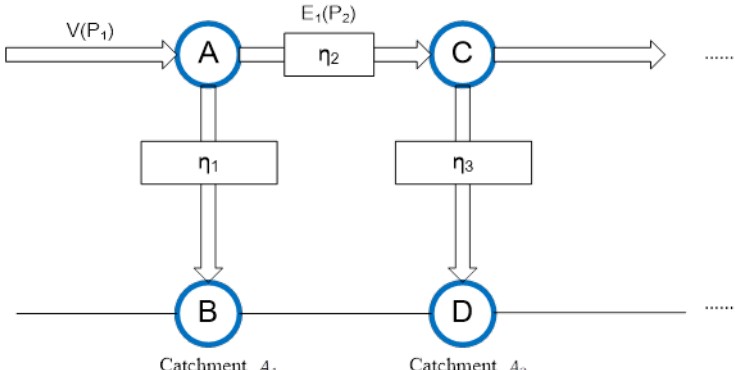

**Figure 2:** Water movement within a catchment and between catchments over a long duration. The figure illustrates the transformation and transportation of the precipitation $P_1$ using a Lagrangian particle tracking method. The arrows represent the path and direction of water movement. Note that Catchment $A_2$ represents a group of catchments in which the water vapor that evaporated from Catchment $A_1$ might precipitate. V is defined as the water vapor that is precipitated onto Catchment $A_1$, which therefore quantitatively equals the precipitation $P_1$ fell down to Catchment $A_1$. The Nodes A and C indicate water in a gas state, while the Nodes B and D indicate water in liquid state. $E_1$ represents the evaporation from Catchment $A_1$, and $P_2$ represents the precipitation from $E_1$.



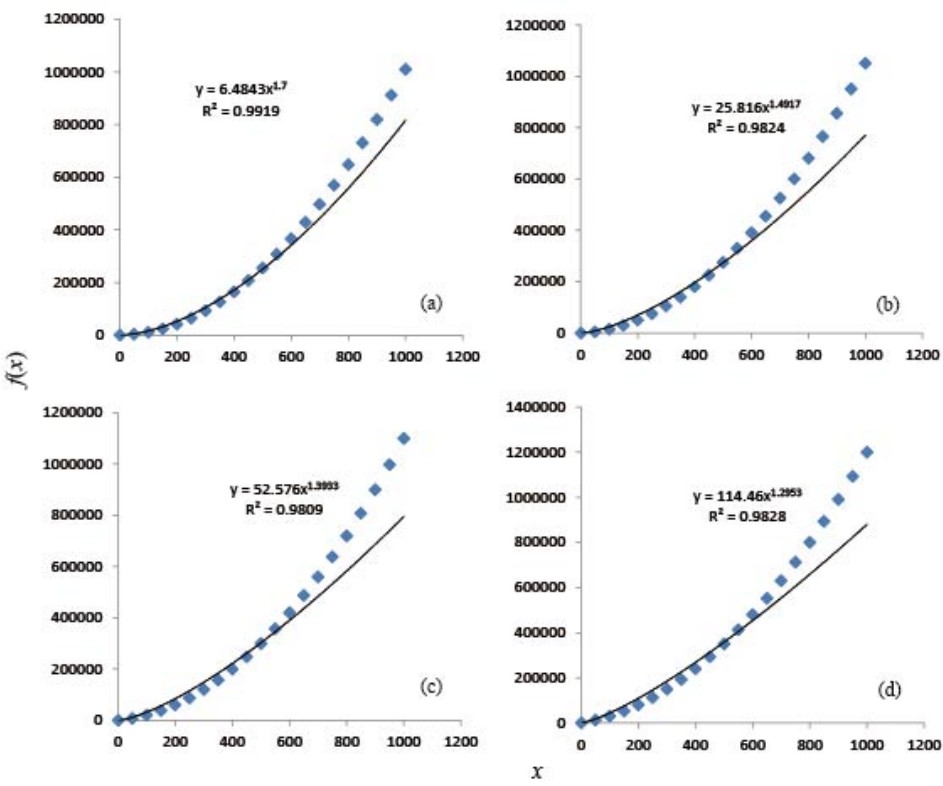

**Figure 3**: Regression analysis on the quadratic polynomial $f(x) = x^2 + bx$ with (a) $b = 10$, (b) $b = 50$, (c) $b = 100$, and (d) $b = 200$ by using power functions.

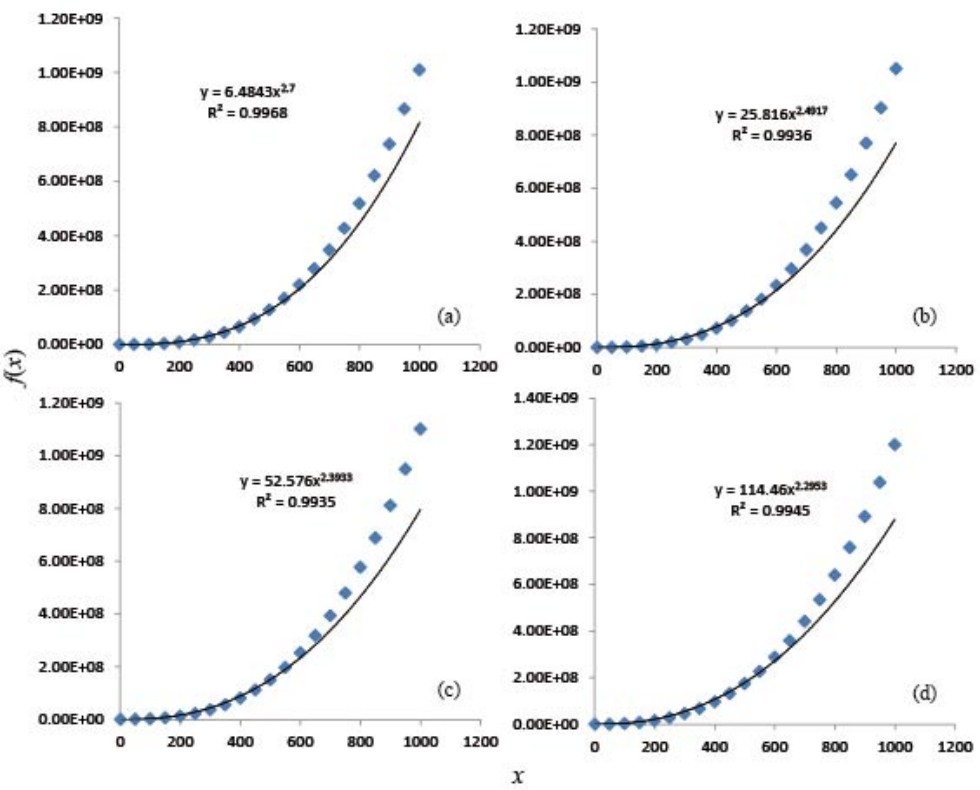

**Figure 4**: Regression analysis on the cubic polynomial $f(x) = x^2 + bx$ with (a) $b = 10$, (b) $b = 50$, (c) $b = 100$, and (d) $b = 200$ by using power functions.



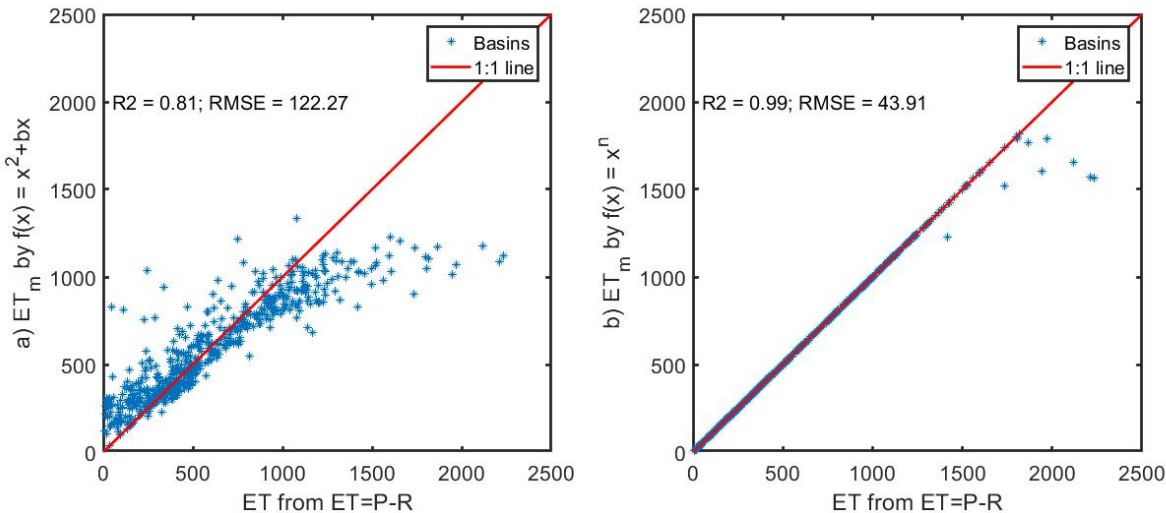

**Figure 5**: Global analysis on the generalized flux function with (a) quadratic function $f(x) = x^2 + bx$ and (b) power function $f(x) = x^n$.