# Peer review of "Towards understanding the mean annual water-energy balance equation based on an Ohms-type approach"

_Hydrology and Earth System Sciences, 2019_

## Referee Comment (RC1) · Anonymous Referee #1 · 11 Aug 2019

The authors aim to define a new Budyko-type relationship (it seems) by using an Ohm's approach.

The article has a very strange structure. There is really no justification on the knowledge gap for the research. The literature review is rather "old" and convoluted, mentioning relevant but rather old formulations. Much has happened in the Budyko field after the Zhang, Fu and MCY formulations. Budyko-type empirical models of ET. I couldn't even find in the text what is the Ohm's law or approach that the authors are discussing. In the beginning, it sounded interesting to be able to include moisture recycling into the Budyko framework, by using a Lagrangian particle tracking method, but

then, in Page 4 to 6 I just got lost. The manuscript is made up of disconnected parts without a clear thread.

You end up with another expression of E in terms of P and Eo before the discussion, the MCY formulation. Is this suppose to be the main result, that your formulations ends up in the same expression? I think the article goes into too much detail, and looses the big picture.

You are giving mathematical explanations and derivations from the beginning, but I don't really understand what is the research question of this manuscript. Which are the objectives? What is the hypothesis? I am not saying that the new formulation is wrong, but rather what is the purpose of it? It might be interesting what you state, but the relevance or how it contributes to the field is not evident.

Other issues:

The abstract and conclusions are complicated and dense to read.

L. 23 what equation, you have not shown it

Eq. 1 and 2 – you are not defining variables in the equations, and Eq. looks strange. E in terms of E?

L: 5 and 6 . Which is the dry condition and which the wet condition?

L. 9 an analytical equation of what?

L. 10 a derivative of the mean water energy balance? that is not variable?

L.11 What is n, what is m? Specially n results being important, since it is shown in the last equation but you never say what it means.

L: 20 but isn't Fus equation also an implicit function?

Some figures could greatly help, showing the Budyko space.

L. 13 I have never her of the Carrot Limit.

In Eq. 9, what is the physical meaning of conductance?

L. 2 what is the Ohms approach? You have not talked about it or introduced it?

Section 2. Still no sign of the Ohms approach.

The structure of the manuscript is really strange. Where are the results?

So the main result do this article is the last equation, which you don't even number. It looks like yet another expression of E in terms of P and Eo. This can have value at some point, but the relevance is not discussed.

As a recommendation, I think that the authors should rewrite the article thinking in a broader audience. Explaining variables and equations much more clearly. Giving a logical structure to the manuscript, and defining a literature review that shows the knowledge gap, and the aim and objectives that will be used to address that knowledge gap. Also, a clearer relationship between the intercatchemnt assessment (Figure 1) and the Budyko framework would increase the value of the manuscript, it is not clear from the current version. Also, what is the reason of using the Lagrangian and Ohm's approaches? It is not stated.

Also, maybe putting some real values for a specific catchment to the conceptualization reached by the authors could make its application easier as well as understanding its advantages.

―――――――――――――――――

---

## Referee Comment (RC2) · Anonymous Referee #2 · 1 Nov 2019

The author try to present a study on explaining the Budyko-framework based on Ohms-law analogy. They use an interesting moisture recycling approach, that made me really enthousiastic in the beginning of the paper. However, the paper is so oddly written that I quickly was not able to understand it anymore. I am completely lost in equations that are not properly explained and the weird structure. Nowhere in the paper I see a justification that Ohms-law can be applied on catchment evaporation. Nowhere I see any validation of the study. Figures 3-5 might be the validation, but I have no clue what I see here. What is 'x'? Nor is clear what this paper adds: what do the authors try to solve (i.e. knowledge gap is missing)? Where are the results? Are that the graphs in Section 4?!?. This section is also totally unclear. It seems partly an introduction, partly

results... at least for sure no 'discussion'.

So to conclude: I am not able to grasp the paper at all. Maybe the method is OK, but based on the weird manuscript structure, it is impossible to follow and to judge it. Please improve the structure and link more to the physical processes in hydrology (HESS is a hydrological journal). Additionally, the language can also be improved.

Specific comments:

-P1 L10: "however, few hydrological processes were involved in the derivation". Is it? And are they included in your approach?

-P1 L13-17: ?? which new constraint? What is a generalized flux?

-P1 L19: What is a homogeneity constraint?

-P2 L10-11: The 'd' of derivative should not be italic. it's not a parameter.

-P3 L6: Be consistent in capital and non-capital parameters (phi)

-P3 L21: "accordingly". I don't get how your objective links to the existing work. Please describe the knowledge gap and the relevance of your work.

-P4 L1: ".. phase transitionS...... namely evaporation AND condensation.."

-P4 L10: It's more that once you consider a time scale of more than 1 year, you can neglect the storage change term.

-P4 L13-33: This part has to be rewritten. I am completely lost here. What is the difference between i and j? Why are the assumptions valid? How do the authors justify that evaporation is driven by a 'potential' difference dU? I do see this analogy with e.g. Darcy's Law where flow is driven by a pressure difference. However, in the case of evaporation it's a trade-off between evaporative power (Epot) and water availability. This is the main idea behind Budyko. So I don't see why Ohms law analogy can be used in the Budyko framework.

[Figure]

-Figure 1-2: how are these figures linked to each other? I think they both try to explain the method, but I don't see how figure 2 follows from figure 1.

- Figure 1: So the authors use a recycling approach, where evaporation from 1 catchment can re-precipitated in the same or another downwind catchment. This is true, but what I am missing is that catchment 2 receives, beside rainfall from evaporated moisture from catchment 1, als rainfall from other catchments. How is this incorporated? Please also read and refer to: van der Ent, R. J., Savenije, H. H. G., Schaefli, B., Steele-Dunne, S. C. (2010). Origin and fate of atmospheric moisture over continents. Water Resources Research, 46(9), W09525. doi.org/10.1029/2010WR009127

- P5 L18: What is the physical meaning of U2-U1?

- Section 2.2: I would start with this section before explaining the mathematical equations. Additionally, section 2.2 should more link to the physics (as the title suggests). Why is it valid to apply this approach??

-P5 L15-22: This list is not introduced

-P5 L23-31: This list is not introduced.

-P6: completely lost here from here onwards.

---

## Author Comment (AC1) · 30 Dec 2019

Response to Anonymous Referee #1

The authors aim to define a new Budyko-type relationship (it seems) by using an Ohm's approach. The article has a very strange structure. There is really no justification on the knowledge gap for the research. The literature review is rather "old" and convoluted, mentioning relevant but rather old formulations. Much has happened in the Budyko field after the Zhang, Fu and MCY formulations. Budyko-type empirical models of ET. I couldn't even find in the text what is the Ohm's law or approach that the authors are discussing. In the beginning, it sounded interesting to be able to include moisture

recycling into the Budyko framework, by using a Lagrangian particle tracking method, but then, in Page 4 to 6 I just got lost. The manuscript is made up of disconnected parts without a clear thread. You end up with another expression of E in terms of P and Eo before the discussion, the MCY formulation. Is this suppose to be the main result, that your formulations ends up in the same expression? I think the article goes into too much detail, and looses the big picture. You are giving mathematical explanations and derivations from the beginning, but I don't really understand what is the research question of this manuscript. Which are the objectives? What is the hypothesis? I am not saying that the new formulation is wrong, but rather what is the purpose of it? It might be interesting what you state, but the relevance or how it contributes to the field is not evident.

Response:

We thank the reviewer for taking the time to review our manuscript and for the invaluable comments. As for the knowledge gap for this research, we think it is that between the mathematical equation and hydrological mechanism and stated it both in the abstract and introduction, i.e. "few hydrological processes were involved in the derivation". To our knowledge, the Fu and MCY formulations have been considered as the analytical ones and have been the most widely used ones, and their equivalence has be proved. Therefore, we focused on the MCY formulation. Previous derivation is filled with mathematical reasoning and the dimensional analysis. There was so few hydrological process involved in previous derivation, except the boundary conditions, that we tried to link it with more hydrological meaning. Although recent studies tried to derive the mean annual water-energy balance based on physical assumptions such as the principle of maximum entropy production and Carnot Limit, as reviewed in our introduction, their results include several not easily accessible and vague parameters, which leads to that they were hardly applied. Therefore, the purpose of our manuscript was stated in the abstract and the introduction: to fill in the gap between the derivation of MCY equation and hydrological processes. As for the "old and convoluted literature review", we

agree with the reviewer that "much has happened in the Budyko field after the Zhang, Fu and MCY formulations", but the Fu and MCY formulations were considered as the analytical ones and have been the most widely used ones, and their equivalence has be proved. In fact, we tried to derive the mean annual water-energy balance base on the Ohm's balance and obtained a formulation same to the MCY equation. Therefore, we focused on the MCY formulation and gave a detail review on it. Regarding "mentioning relevant but rather old formulations", this paper aims at the derivation of mean annual water-energy balance, so we have to focus on the "old and convoluted" equations before Zhang, Fu and MCY formulations. By tracking those equations, we get much more understanding of the derivation. As for the Ohm's law, I am sorry that we didn't describe that clearly. We will give more detailed explanation in the revised version. As well known, at a long time scale, the water evaporated into atmosphere will be precipitated on land due to water cycle. In our manuscript, we defined the catchment network for water (vapor) transformation and transportation. To define the catchment network, we track the water movement using Lagrangian particle tracking method, i.e. we took the water precipitated into the first catchment as research object and marked it as P1; we focuses on the subsequent transportation and transformation of P1 and all the catchments that the water enters into was defined as the catchment network. Remarkably, for a special catchment of the catchment network, part of precipitation comes from P0 and the rest comes from other sources, and we only studied the former. In Figure 1, Catchments A2,j (j=1, 2, 3, . . . ) represent all the catchments that the evaporated water from Catchment 1 can fall down with a form of precipitation, where water vapor has been through once of evaporation-precipitation process from P1; while Catchments A3,j (j=1, 2, 3, . . . ) represent the catchments that the evaporated water from Catchments A2,j (j=1, 2, 3, . . . ) can fall down with a form of precipitation, where water vapor has been through twice of evaporation-precipitation process from P1. Regarding the precipitation in Catchment 2 as the reviewer concerns, some comes from Catchment 1 and the rest comes from other sources; however, according to the Lagrangian particle tracking method, we only focus on the part from Catchment 1. Also,

we can establish the balance equation of only the water from Catchment 1 for Catchment 2, $P2 = E2 + R2$, where P2 being the precipitation from Catchment 1, E2 and R2 being the evaporation and the runoff from the evaporated water from Catchment 1, respectively.

Further remarks:

abstract: "The abstract and conclusions are complicated and dense to read.

Response: 'Thank very much for your comment and we will add some detailed explanations in the revised version.

L. 23 what equation, you have not shown it

Response:

This equation represents the mean annual water-energy balance equation mentioned in L. 22.

Eq. 1 and 2 – you are not defining variables in the equations, and Eq. looks strange. E in terms of E?

Response:

They were defined in L. 22 and L. 23. E in terms of E is just a mathematical form.

L: 5 and 6 . Which is the dry condition and which the wet condition?

Response:

Respectively, ðİŘÿ → ðİŚČ as ðİŘÿÂň0 → ∞ is dry condition, i.e., evaporation approaches precipitation when potential evaporation is large enough, and ðİŘÿ → ðİŘÿ0 as ðİŚČ → ∞ is wet condition, i.e., evaporation approaches potential evaporation when precipitation is large enough.

L. 9 an analytical equation of what?

[Figure]

Response:

An analytical equation of the Budyko hypothesis. We will revise the expression to make it more clear.

L. 10 a derivative of the mean water energy balance? that is not variable?

Response:

I am sorry for the inaccurate expression, and it should be a derivative of E with respect to precipitation. We will revise it.

L.11 What is n, what is m? Specially n results being important, since it is shown in the last equation but you never say what it means.

Response:

n is a parameter introduced to reflect the characteristics of underlying surface by Bagrov (1953) and m is a variable defined as m=(n+1)/n by Mezentsev (1955) to integrate the derivative. We will revise the expression in the resubmitted version.

L: 20 but isn't Fus equation also an implicit function? Some figures could greatly help, showing the Budyko space.

Response:

Yes. Fu's equation is an implicit function of E0 − E and P (or P − E and E0), stated in L.14. Thank you for your suggestion, and we will add figures to show the difference between the two equations.

L. 13 I have never her of the Carrot Limit.

Response:

I am sorry for our carelessness. It's a typo. It should be "Carnot Limit". I am sorry for our carelessness and will revise it in the revised version.

In Eq. 9, what is the physical meaning of conductance?

Response:

I am sorry for our carelessness, and it should be flux.

L. 2 what is the Ohms approach? You have not talked about it or introduced it?

Response:

As for the Ohm's law, I am sorry that we didn't describe that clearly. The Ohm's law can be referred as Equation (11), i.e. the flux can be expressed as the function of the wate potential difference and the resistance of the water vapor movement or transportation process. We will give more detailed explanation in the revised version. As well known, at a long time scale, the water evaporated into atmosphere will be precipitated on land due to water cycle. In our manuscript, we defined the catchment network for water (vapor) transformation and transportation. To define the catchment network, we track the water movement using Lagrangian particle tracking method, i.e. we took the water precipitated into the first catchment as research object and marked it as P1; we focuses on the subsequent transportation and transformation of P1 and all the catchments that the water enters into was defined as the catchment network. Remarkably, for a special catchment of the catchment network, part of precipitation comes from P0 and the rest comes from other sources, and we only studied the former. In Figure 1, Catchments A2,j (j=1, 2, 3, . . . ) represent all the catchments that the evaporated water from Catchment 1 can fall down with a form of precipitation, where water vapor has been through once of evaporation-precipitation process from P1; while Catchments A3,j (j=1, 2, 3, . . . ) represent the catchments that the evaporated water from Catchments A2,j (j=1, 2, 3, . . . ) can fall down with a form of precipitation, where water vapor has been through twice of evaporation-precipitation process from P1. Regarding the precipitation in Catchment 2 as the reviewer concerns, some comes from Catchment 1 and the rest comes from other sources; however, according to the Lagrangian particle tracking method, we only focus on the part from Catchment 1. Also, we can establish the

balance equation of only the water from Catchment 1 for Catchment 2, P2 = E2 + R2, where P2 being the precipitation from Catchment 1, E2 and R2 being the evaporation and the runoff from the evaporated water from Catchment 1, respectively.

The structure of the manuscript is really strange. Where are the results?

Response:

I am sorry for the unclear structure. This manuscript aimed to derive the mean annual water-energy balance equation, so the results are Section 2. And we will revise the expression in the resubmitted version.

Please also note the supplement to this comment:
https://www.hydrol-earth-syst-sci-discuss.net/hess-2019-283/hess-2019-283-AC1-supplement.pdf

---

## Author Comment (AC2) · 30 Dec 2019

Response to Anonymous Referee #2

The author try to present a study on explaining the Budyko-framework based on Ohm-slaw analogy. They use an interesting moisture recycling approach, that made me really enthousiastic in the beginning of the paper. However, the paper is so oddly written that I quickly was not able to understand it anymore. I am completely lost in equations that are not properly explained and the weird structure. Nowhere in the paper I see a justification that Ohms-law can be applied on catchment evaporation. Nowhere I see any validation of the study. Figures 3-5 might be the validation, but I have no clue what

I see here. What is 'x'? Nor is clear what this paper adds: what do the authors try to solve (i.e. knowledge gap is missing)? Where are the results? Are that the graphs in Section 4?!?. This section is also totally unclear. It seems partly an introduction, partly results... at least for sure no 'discussion'. So to conclude: I am not able to grasp the paper at all. Maybe the method is OK, but based on the weird manuscript structure, it is impossible to follow and to judge it. Please improve the structure and link more to the physical processes in hydrology (HESS is a hydrological journal). Additionally, the language can also be improved.

Response:

We really appreciate the reviewer taking time to review our manuscript and giving very important comments. At the same time, I want to say sorry for our carelessness that we forgot to update the order number of sections after removing Section 3 in the initial version. In this version, the result section is Section 2, i.e. Ohms-type approach. We will carefully check and do a throughout revision in the resubmission.

Regarding to "Ohms-law approach", it was described as Equation (11), i.e. the flux can be estimated as the function of the potential difference and the resistance of the water vapor movement or transportation process. And we will add more explanations to our manuscript.

Regarding to the structure of this paper, we will add a chart to explain the physical variables used in this manuscript. In addition, the result of this manuscript

Regarding to the validation, our logic is that our derived formulation based on totally different derivation but converge to the widely-used MCY equation, and the MCY validated in many previous studies means that our result can be validated.

Further remarks:

P1 L10: "however, few hydrological processes were involved in the derivation". Is it? And are they included in your approach?

Response: I am sorry for our vague expression. Compared with purely mathematic transformation in previous studies, our derivation tried to reason it by using an Ohm-type approach based on the water transportation between catchments.

-P1 L13-17: ?? which new constraint? What is a generalized flux?

Response: The new constraint is $1/(f(E))=1/(f(E\_0))+1/(f(P))$, and the general flux is explained in Section 2, i.e. "the generalized flux is defined as the potential difference divided by the resistance and is a function of flux".

-P1 L19: What is a homogeneity constraint?

Response: This constraint is explained in previous derivation Yang et al., 2008. i.e., the generalized function has the same form for both vapor transportation and chase transition, and in other words, precipitation and potential evaporation have an equalized effect on evaporation. We will add more explanation on it in the revised version.

-P2 L10-11: The 'd' of derivative should not be italic. it's not a parameter -P3 L6: Be consistent in capital and non-capital parameters (phi)

Response: I am sorry for our carelessness and thank you for your specific comments. And we will revise our manuscript.

-P3 L21: "accordingly". I don't get how your objective links to the existing work. Please describe the knowledge gap and the relevance of your work.

Response: The knowledge gap is that previous derivations lack the hydrological meaning but be full of dimensional analysis and mathematical assumptions. That's why we want to add more physical or hydrological meaning to Budyko equation. We will revise this part to make the logic more clear.

-P4 L1: ".. phase transitionS...... namely evaporation AND condensation.."

Response: Thanks a lot for your specific comments and we will revise our manuscript.

-P4 L10: It's more that once you consider a time scale of more than 1 year, you can neglect the storage change term.

Response: Yes, we did the reasoning on the time scale of more than 1 year. And we will improve the expressions.

-P4 L13-33: This part has to be rewritten. I am completely lost here. What is the difference between i and j? Why are the assumptions valid? How do the authors justify that evaporation is driven by a 'potential' difference dU? I do see this analogy with e.g. Darcy's Law where flow is driven by a pressure difference. However, in the case of evaporation it's a trade-off between evaporative power (Epot) and water availability. This is the main idea behind Budyko. So I don't see why Ohms law analogy can be used in the Budyko framework.

Response: Thank you very much for your invaluable comment. We will rewrite this part and add more explanations. i and j have different meaning, i.e., one group of catchment is denoted as i, and different catchments within this group are denoted as different j. We agree with you that evaporation is a trade-off between evaporative power and water availability, but at the same time, water vapor above one catchment will be divided into two parts, one forming runoff (by precipitating) and the one transported outside of the catchment over a long time scale, and the Ohms analogy was used to quantify the two parts. Furthermore, we can obtain the MCY equation, which is a widely-used equation for the Budyko hypothesis.

-Figure 1-2: how are these figures linked to each other? I think they both try to explain the method, but I don't see how figure 2 follows from figure 1.

Response: Figure 1 characterizes the water transportation between different catchments while Figure 2 only denotes the water transportation from first catchment group to the second one. In addition, Figure 2 introduces resistances to quantify the fluxes, water transportation from one catchment group to another one and the net flux (P - E).

[Figure]

- Figure 1: So the authors use a recycling approach, where evaporation from 1 catchment can re-precipitated in the same or another downwind catchment. This is true, but what I am missing is that catchment 2 receives, beside rainfall from evaporated moisture from catchment 1, als rainfall from other catchments. How is this incorporated? Please also read and refer to: van der Ent, R. J., Savenije, H. H. G., Schaefli, B., Steele-Dunne, S. C. (2010). Origin and fate of atmospheric moisture over continents. Water Resources Research, 46(9), W09525. doi.org/10.1029/2010WR009127

Response: Thank you for your comment and suggestion. In this manuscript, we just track the precipitation in Catchment 1 using a Lagrangian approach. Regarding the precipitation on Catchment, we agree with you that some from evaporated moisture from other catchments, but we only track the part from evaporated moisture from Catchment 1. In another word, it is only the transformation and transportation of the initial evaporation from Catchment 1 that is shown in Figure 1.

- P5 L18: What is the physical meaning of U2-U1?

Response: It is the water potential difference between A and C.

-P5 L15-22: This list is not introduced -P5 L23-31: This list is not introduced.

Response: P5 L15-22 are the description on the water potential and flux in a catchment network, while P5 L23-31 are the conclusion the reasoning from P5 L15-22 based on the Ohms approach.

Please also note the supplement to this comment:
https://www.hydrol-earth-syst-sci-discuss.net/hess-2019-283/hess-2019-283-AC2-supplement.pdf

———————————————